# Bisphenol A Exposure Interferes with Reproductive Hormones and Decreases Sperm Counts: A Systematic Review and Meta-Analysis of Epidemiological Studies

**DOI:** 10.3390/toxics12040294

**Published:** 2024-04-17

**Authors:** Lei Lü, Yuan Liu, Yuhong Yang, Jinxing He, Lulu Luo, Shanbin Chen, Hanzhu Xing

**Affiliations:** 1School of Food Science and Engineering, Qilu University of Technology, Shandong Academy of Sciences, Jinan 250353, China; lvlei831005@163.com (L.L.); ly85951005@163.com (Y.L.); yyhforever1108@163.com (Y.Y.); jinhe@qlu.edu.cn (J.H.); 15662758306@163.com (L.L.); 2Institute of Food & Nutrition Science and Technology, Shandong Academy of Agricultural Sciences, Jinan 250100, China

**Keywords:** endocrine disruptors, male fertility, reproductive hormones, risk of below-reference sperm quality, sperm counts, bisphenol A

## Abstract

Bisphenol A (BPA), an acknowledged endocrine disrupter, is easily exposed to humans via food packaging and container. However, a consensus has not been reached on the extent to which BPA exposure affects the reproductive system. We therefore conducted this systematic review and meta-analysis to elucidate the relationship between BPA exposure and male reproduction-related indicators. Up to October 2023, a comprehensive search was carried out in the PubMed, Embase, Cochrane and Web of Science, and 18 studies were ultimately included. β coefficients from multivariate linear regression analyses were pooled using a random effects model. The results showed that the urinary BPA concentration was negatively correlated with the sperm concentration (β coefficient = −0.03; 95% CI: −0.06 to −0.01; I^2^ = 0.0%, *p* = 0.003) and total sperm count (β coefficient = −0.05; 95% CI: −0.08 to −0.02; I^2^ = 0.0%, *p* < 0.001). In addition, BPA concentrations were associated with increased sex hormone-binding globulin (SHBG) levels, increased estradiol (E_2_) levels, and reduced biologically active androgen levels. However, the relationship between an increased risk of below-reference sperm quality and BPA exposure was not robust. This systematic review revealed that BPA exposure disrupts reproductive hormones, reduces sperm counts and may ultimately adversely affect male reproduction.

## 1. Introduction

The declining fertility and increasing population aging are global challenges. A number of studies have reported a rising prevalence of infertility with a worsening global fertility rate over the last few decades [1]. A reduced quality of semen caused by physiological and psychological factors individually or in combination have substantially contributed to this circumstance [1,2,3]. Research shows that there are more than 40 million individuals experiencing infertility in China, causing considerable social and economic impacts [4]. The reproductive health of the infertile population has become a public health issue of worldwide concern [5].

Endocrine disruptors are key factors that negatively affect semen quality, and BPA is the most common endocrine disruptor [6]. The structure of bisphenol A (Figure 1) is similar to that of endogenous E_2_ and researchers have proved its affinity to estrogen receptors [6]. In addition, BPA interferes with the hypothalamic–pituitary–testicular axis where these three glands act in concert to control development, reproduction and aging. Gonadotropin-releasing hormone is secreted from the hypothalamus and stimulate the anterior portion of the pituitary gland to produce gonadotropin luteinizing hormone (LH) and follicle-stimulating hormone (FSH). In response to LH and FSH, the testis produces sex hormone testosterone (T) and inhibits the hypothalamus or pituitary gland in return. It was reported that BPA interferes with the synthesis of the FSH, LH and T levels, and influences the functional cellular effects of the testes [5,6]. By mimicking the effects of estrogen and interfering with its regulation, BPA exposure can block or alter spermatogenesis, negatively affecting sperm production and quality [7,8].

BPA is widely used in the production of materials such as polycarbonate, epoxy resins, polysulfone resins and many other polymeric materials [9]. With the widespread use of plastic products, the general population is being increasingly exposed to BPA. It was reported that the median BPA concentration can reach 4.2 μg/g creatinine in urine, even reaching 685.9 μg/g creatinine (Cr) in occupationally exposed workers [8]. According to the European Food Safety Authority, the tolerable daily intake of BPA was reduced from 4 μg/kg bw per day to 0.2 ng/kg bw per day [10].

The toxicity of BPA is well established at a relatively high dose [10]. At present, measurements have been taken to control BPA exposure in affected countries [5,6]. However, whether unavoidable sustained BPA exposure at low doses contributes to reduced male fertility has not been determined. Long-standing observational studies have attempted to uncover this problem, but the results have been inconsistent. In Mendiola’s study, BPA was found to have no effect on semen quality parameters [11]. In Li’s study, men with detectable BPA in their urine had a multiple times greater risk of having a reduced sperm concentration, lower sperm motility and decreased sperm count than did healthy men [12].

To date, several reviews and meta-analyses have explored the relationship between BPA and male reproduction. Rodent animals were most frequently adopted to investigate the toxicity of BPA. Two meta-analyses examined changes in sperm quality and reproductive organ weights in BPA-exposed male rodents [13,14]. Some studies have discussed the effects of BPA on reproductive hormones in animals, including T and E_2_ [13,15,16,17]. Meanwhile, whether these conclusion from animal studies are appropriate for humans needs to be verified. Kortenkamp carried out a systematic review of epidemiological studies and attributed the divergent findings on sperm quality to differences in exposure conditions [18]. Castellini meta-analyzed the correlation between BPA and semen quality [19]. However, findings of reproductive hormones from epidemiologic studies are not well discussed. Therefore, this study comprehensively reviewed the epidemiological studies investigating male reproduction toxicity of BPA exposure in a qualitative and quantitative analysis combined manner. Importantly, not only sperm parameters but also male reproductive hormones and risk of below-reference sperm quality were discussed.

## 2. Materials and Methods

### 2.1. Search Strategy

A systematic search was performed in Embase, the Cochrane Library and PubMed for relevant English-language papers published up to October 2023 that focused on the association between BPA exposure and the male reproductive system. Medical subject headings terms were developed by combining subject and free words with “AND” or “OR”: “Bisphenol A”, “male reproduction”, “sperm parameters” and “reproductive hormones” (refer to Appendix A).

### 2.2. Study Screening

Two researchers independently reviewed and screened the qualified studies in two stages. The first stage involved reviewing the title and abstract, and the second involved screening the full text. Disagreements were resolved promptly through discussion. The outcome was the relationship between the BPA concentration and male reproduction parameters, and articles that fulfilled the criteria were included in the systematic review and meta-analysis. The inclusion criteria were as follows: (1) case–control or observational studies, including cross-sectional studies and cohort studies; (2) studies that evaluated BPA exposure; and (3) studies that evaluated the association between BPA levels and reproductive hormones, semen parameters or the risk of below-reference sperm quality.

The exclusion criteria were as follows: (1) duplicate publications; (2) conference proceedings, reviews, and book chapters; (3) in vitro studies and animal model studies; and (4) studies with incomplete data or information unrelated to semen parameters and reproductive hormones.

### 2.3. Data Extraction

Data on the general characteristics of the eligible studies were extracted, including the first author, year of publication, country, study design, and type of study population, multivariate linear regression analysis (β coefficients with 95% CI) results of the association between urinary BPA levels and semen parameters and confounding factors used in regression model adjustment.

### 2.4. Quality Assessment

The Newcastle–Ottawa Scale (NOS) was used for quality assessment. According to the NOS, a study can be awarded a maximum of 9 stars. If a study fulfills every aspect of the following items, it receives one star. Four stars are set for sample selection, including representativeness of the exposed cohort, selection of the non-exposed cohort, ascertainment of exposure and no outcomes of interest at first. Two stars can be awarded from comparability of whether the study was controlled for confounding factors. And 3 stars were set for assessment of outcome, adequacy of follow-up length and adequacy of follow-up cohorts. In the current meta-analysis, studies with a quality score of 7 or higher were considered “high quality”, while those with a quality score of 6 or below were considered “low quality”. Two researchers (Y.L. and S.C.) independently assessed the quality. When two researchers hold inconsistent opinions about an article, the decision of the third researcher (H.X.) is conclusive.

### 2.5. Statistical Analysis

The adjusted coefficient (95% CI) of the multivariate linear regression model was used to calculate the overall effect size. Cochran’s Q test and the I^2^ test were used to examine the heterogeneity among the studies. Publication bias was investigated by funnel plot and Begg’s test. Statistical analysis was performed using STATA version 16.0. A *p* value < 0.05 was considered to indicate statistical significance.

## 3. Results

### 3.1. Study Characteristics

From the database and online searches, we recovered a total of 162 articles (Figure 2). After the deletion of duplicate articles, 110 articles were identified. Subsequently, we screened and evaluated the articles based on their titles and abstracts, and 76 of them were subjected to full-text review. Ultimately, 18 articles satisfied the criteria and were included in the analysis.

These articles were published between 2010 and 2023. The majority of the articles presented cross-sectional studies, partly with cohort designs. Among the eligible studies, five articles were from China, four were from the USA, two were from Denmark, two were from Spain and two were from Italy; the rest were published in Mexico, Poland and Slovenia. The quality assessment of the articles was performed as shown in Table 1. Based on the Newcastle–Ottawa Scale, only 2 studies received a score of 9, while the remaining 16 articles received a score of 7; all of these studies were considered to be of high quality.

### 3.2. Associations between BPA Exposure and Male Reproductive Hormone Levels

A total of nine articles with 2961 participants reported the association between BPA exposure and sex hormone levels. Five of these studies had a cross-sectional design [8,11,20,23,24], two were cohort studies [21,26], one was a case–control study [22] and one did not specify the study design [25]. Two cohort studies focused on the effect of BPA exposure on boys; one article reported the association in workers occupationally exposed to BPA, and the remaining reported the effect on male adults in the general population. The urinary BPA concentration in occupationally exposed workers reached 3671.8 μg/g Cr, while the urinary BPA concentration in the general population was less than 15.9 μg/g Cr [8].

The associations between BPA levels and male reproductive hormone levels are summarized in Table 2. As heterogeneity was substantially unavoidable, the results were reviewed but not meta-analyzed. A positive correlation between BPA exposure and SHBG or E_2_ levels was most frequently observed [8,11,21,22,23]. The correlation coefficient between urinary BPA and SHBG levels ranged from 0.0293 to 0.07, while that between urinary BPA and E_2_ levels ranged from 0.0362 to 0.49 [8,11]. Increased T levels were reported to be associated with BPA levels [11,24,26]. However, the biologically active free androgen level decreased, as indicated by the negative association coefficient of the free androgen index, free T level, or androstenedione level, which might result from the increased ability of SHBG for binding T [8,11,12]. Adoamnei and Lassen reported a positive correlation between BPA and LH levels [20,25]. However, another trophic hormone, FSH, was found to be negatively correlated with BPA exposure [8,20].

### 3.3. Associations between BPA Exposure and Male Sperm Parameters

In total, eight studies with 2468 participants reported associations between BPA exposure and sperm parameters, which included sperm concentration, total sperm count, sperm motility, semen volume and normal sperm morphology [11,20,25,27,28,29,30,31]. Of these, four were cross-sectional studies [11,20,29,30], two were cohort studies [27,28] and the remaining two did not describe the study design [25,31]. The meta-analysis was conducted according to the β coefficients and 95% confidence intervals of the included studies. The characteristics of these studies are summarized in Table 3.

A comprehensive estimation revealed a negative interaction between the urinary BPA concentration and sperm concentration (β coefficient = −0.03; 95% CI: −0.06 to −0.01; I^2^ = 0.0%, *p* = 0.003; Figure 3a) and total sperm count (β coefficient = −0.05; 95% CI: −0.08 to −0.02; I^2^ = 0.0%, *p* < 0.001; Figure 3b), with no relationship with sperm motility (β coefficient = −0.40; 95% CI: −0.80 to 0.01; I^2^ = 46.1%, *p* = 0.053; Figure 3c), semen volume (β coefficient = 0.01; 95% CI: −0.02 to 0.05; I^2^ = 67.0%, *p* = 0.466; Figure 3d) or normal sperm morphology (β coefficient = −0.01; 95% CI: −0.03 to 0.01; I^2^ = 0.0%, *p* = 0.269; Figure 3e). Publication bias was accessed with funnel plots. No publication bias was found, as symmetry was observed in funnel plots for sperm concentration, total sperm count, volume and normal sperm morphology, which was also indicated by Egger’s test (*p* = 0.915, 0.790, 0.600, 0.879, respectively), while the funnel plot was asymmetry for analysis of sperm motility (*p* = 0.042). After the trim and fill analysis, still no significant correlation was found between BPA exposure and sperm motility (refer to Appendix A).

### 3.4. Associations between BPA Exposure and the Risk of Below-Reference Sperm Quality

As shown in Table 4, based on four epidemiological studies with 1891 participants, an association was found between exposure to bisphenol A and the risk of sperm parameters deviating from normal values, which may be associated with reproductive disorders. Three of those studies had cross-sectional designs. Two of the cross-sectional study populations were from infertility clinics [32,33], and one was from a young adult population without known infertility [35]. The remaining cohort study focused on men aged 18–40 years without known subfertility [34]. The adjusted mean urinary creatinine concentration of BPA was 1.81 μg/g [33], with a median urinary BPA concentration of 1.30 ng/mL in both studies [32,35].

## 4. Discussion

The present meta-analysis comprehensively explored whether and to what extent environmental exposure to BPA might be associated with male fertility. We included epidemiological studies that used multivariate linear regression models to quantify the association between BPA levels and reproductive hormone levels or semen quality.

Overall, the results showed that BPA exposure was significantly correlated with sex hormone disturbances since BPA is regarded as a well-known endocrine-disrupting chemical [5,6]. Its effect on sex hormones has been well established by preclinical studies [21,22,26]. BPA disrupts the hormone environment in Leydig cells, where T is produced via steroidogenic gene activation through the JNK/c-Jun signaling pathway [36]. Animal studies have also shown that BPA reduces testicular T synthesis by disrupting the normal expression of StAR, a gene encoding a protein that acutely regulates the synthesis of steroid hormones [37,38]. Exposure to BPA disrupts the hypothalamic–pituitary–gonadal axis and decreases the concentrations of LH and FSH, causing a decline in sperm function and sperm production [39]. By reviewing the epidemiological data from men, we noted that SHBG, E_2_ and biologically active androgen were most frequently related to BPA exposure. The concentration of SHBG, which is primarily bound to estrogens and androgens, is regulated by sex hormones. Estrogen stimulates SHBG production, while T inhibits it. BPA itself is structurally similar to E_2_ and has weak estrogenic activity. By acting as an estrogen-like hormone, BPA may competitively bind to SHBG, displacing the endogenous estrogen bound to SHBG, resulting in an increase in the serum E_2_ concentration [40]. Because of the increase in SHBG, the balance between androgen and estrogen in the serum is disturbed, the amount of T that binds to it increases, the androgen that is actually active decreases and spermatogenesis is impaired [41]. Studies have also shown that elevated serum SHBG levels are associated with lower sperm concentrations [42].

The meta-analysis indicated that BPA exposure had a negative effect on sperm concentration and total sperm count. The sperm count is a common indicator for evaluating the male reproductive system. A decreased sperm count is one of the main manifestations of abnormal sperm quality and is relevant to male infertility [1,14]. In contrast, BPA exposure was not associated with a decrease in sperm motility. Collectively, these results suggest that BPA likely interferes with spermatogenesis but does not affect other aspects of sperm quality. BPA exposure reportedly results in degeneration and a reduction in the number of testicular Sertoli cells, which play a crucial role in sperm production and maturation [43,44,45]. Several studies have shown that BPA induced damage to Sertoli cells [43,46]. Zhang suggested that BPA causes dysregulation of the expression of proteins secreted by Sertoli cells during spermatogenesis, which caused direct damage to Sertoli cells [47]. The possible action may involve its estrogenic activity and effects on the level of reproduction-related hormones through the neuroendocrine mechanism of the hypothalamic–pituitary–gonadal pathway [39]. BPA induces apoptotic cell death by affecting the mitochondrial membrane potential in Sertoli cells and increasing the production of reactive oxygen species [46]. In BPA-treated rats, the arrangement of germ cells in the lumen of the fine tubules was disorganized to different degrees, the number of Sertoli cells around the germ cells was low and the spermatogenic capacity was reduced [43,48]. Moreover, BPA penetrates the blood–testis barrier and interferes with sperm production and development, leading to a reduction in sperm count and lower sperm quality [49].

Several studies have also investigated the association between BPA exposure and the risk of below-reference sperm quality. Although no strong evidence indicates that higher BPA exposure increases the risk of below-reference sperm quality, it is worth noting that the BPA level found in these studies was controlled for. In addition, data from infertility clinics and from men with higher BPA levels were more likely to reveal an elevated risk [32,33]. Considering the association between BPA exposure and reproductive hormone disruption or a decrease in sperm count, it was reasonable to speculate that there was a synergistic effect between BPA exposure and other factors that accelerated the subfecundity epidemic in recent decades.

The current meta-analysis has several limitations. First, when performing the meta-analysis, the populations included in the studies were different. Some of the participants were from the general population, some were from hospital fertility centers and some were from BPA-exposure factories. Due to the specificity of the participants, the latter two groups may have a greater proportion of infertile people than the general population. Moreover, a single measurement of semen samples may not reflect accurate individual exposure assessments. Because the body has its own metabolic functions, the correlation between BPA exposure and male semen quality may change at different timepoints. Finally, there are other unmeasured environmental contaminants that may contribute to biased observations. People are exposed to multiple endocrine disruptors simultaneously in their daily environment. It is not possible to determine whether substances in addition to BPA are toxic for reproduction or whether they exacerbate or diminish the effects of BPA.

## 5. Conclusions

In the present review and meta-analysis, evidences from epidemiological studies suggested that BPA, an endocrine disruptor, caused hormone disruption by raising SHBG and E_2_ levels. Moreover, exposure to BPA reduces sperm concentration and sperm counts, but not sperm motility. More studies are needed to further investigate the relationship between BPA exposure and declining male fertility.

## Figures and Tables

**Figure 1 toxics-12-00294-f001:**
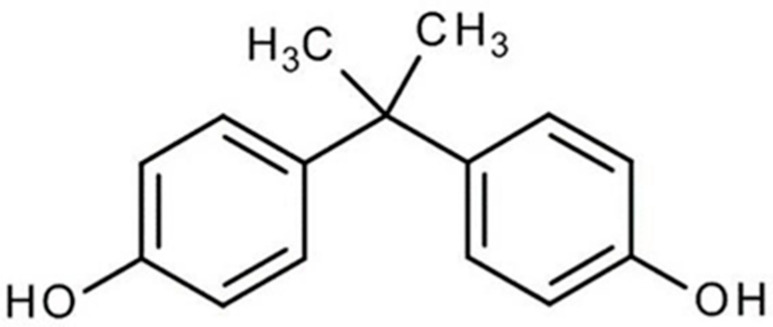
The chemical structure of bisphenol A.

**Figure 2 toxics-12-00294-f002:**
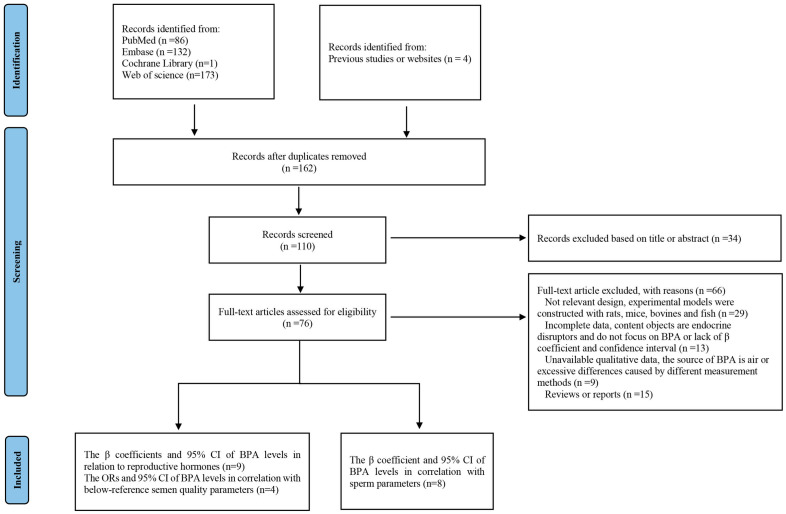
Flow diagram showing the selection process of the study.

**Figure 3 toxics-12-00294-f003:**
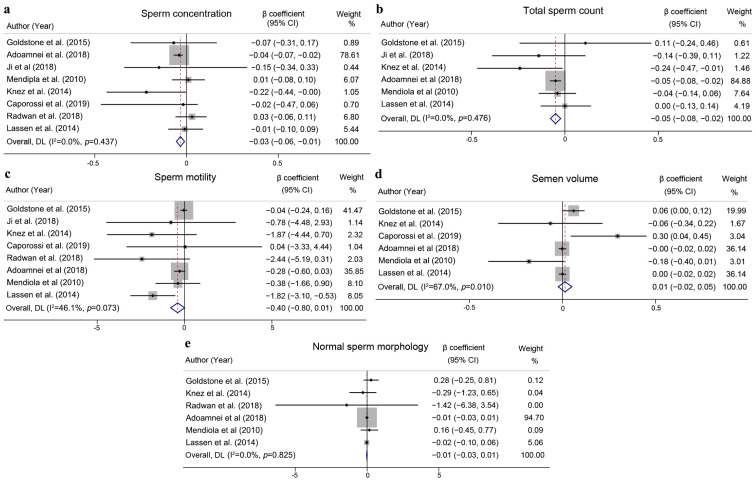
Forest plots pooling the β coefficient of urine BPA levels in correlation with sperm concentration (**a**), total sperm count (**b**), sperm motility (**c**), semen volume (**d**) and normal sperm morphology (**e**). The β coefficient and its 95% confidence intervals of each study was represented as black dot and lateral tips of the horizon line. The size of gray squares indicated proportional to weights. The pooled β coefficient and its 95% confidence intervals was represented as the red dot line and lateral tips of diamond [11,20,25,27,28,29,30,31].

**Table 1 toxics-12-00294-t001:** Quality of the included studies assessed by Newcastle–Ottawa Scale.

Reference	Representativeness of the Exposed Cohort	Selection of the Non-Exposed Cohort	Ascertainment of Exposure	Demonstration That Outcome of Interest Was Not Present at Start of Study	Comparability of Cohorts on the Basis of the Design or Analysis	Assessment of Outcome	Follow-Up Long Enough for Outcomes to Occur	Adequacy of Follow-Up of Cohorts	Total
Adoamnei et al. (2018) [20]	★	★	★	★	★★	★	★	★	9
Mendiola et al. (2010) [11]	☆	☆	★	★	★★	★	★	★	7
Ferguson et al. (2014) [21]	☆	☆	★	★	★★	★	★	★	7
Xiong et al. (2015) [22]	☆	☆	★	★	★★	★	★	★	7
Zhou et al. (2013) [23]	★	★	★	★	★★	★	★	★	7
Galloway et al. (2010) [24]	☆	☆	★	★	★★	★	★	★	7
Lassen et al. (2014) [25]	☆	☆	★	★	★★	★	★	★	9
Mustieles et al. (2017) [26]	☆	☆	★	★	★★	★	★	★	7
Liu et al. (2015) [8]	☆	☆	★	★	★★	★	★	★	7
Goldstone et al. (2015) [27]	☆	☆	★	★	★★	★	★	★	7
Knez et al. (2014) [28]	☆	☆	★	★	★★	★	★	★	7
Ji et al. (2018) [29]	☆	☆	★	★	★★	★	★	★	7
Caporossi et al. (2019) [30]	☆	☆	★	★	★★	★	★	★	7
Radwan et al. (2018) [31]	☆	☆	★	★	★★	★	★	★	7
Meeker et al. (2010) [32]	☆	☆	★	★	★★	★	★	★	7
Chen et al. (2022) [33]	☆	☆	★	★	★★	★	★	★	7
Pollard et al. (2019) [34]	☆	☆	★	★	★★	★	★	★	7
Benson et al. (2021) [35]	☆	☆	★	★	★★	★	★	★	7

★ represent one score; ☆ represents zero score.

**Table 2 toxics-12-00294-t002:** Studies examining the relationship between bisphenol A exposure and male reproductive hormones.

Reference	Study Design	Country	BPA Concentration	Age	No. Participants	Population Type	Main Outcome(s)	Adjustments
Adoamnei et al. (2018) [20]	Cross-sectional study	Spain	2.3 (0.16–11.5) ^a^ ng/mL	20.4 (18.1–22.8) ^a^	215	Healthy young university students	A significant positive association between urinary BPA concentrations and LH levels (β = 0.07, 95% CI: 0.02 to 0.12, *p* < 0.01).	Body mass index (BMI), smoking status, presence of varicocele, urinary creatinine concentration, ejaculation abstinence time and time to start of semen analysis.
Mendiola et al. (2010) [11]	Cross-sectional study	USA	1.5 (<LOD-6.5) ^b^ g/mL	NA ^c^	302	Male partners of pregnant women	A significant inverse association between urinary BPA concentration and FAI levels (β = −0.05, 95% CI: −0.09 to −0.004), as well as a significant positive association between BPA and SHBG (β = 0.07, 95% CI, 0.007 to 0.13).	Age, age squared, BMI, study center, stressful life events and ejaculation abstinence time.
Ferguson et al. (2014) [21]	Cohort study	Mexico	NA ^c^	8.10–14.4 ^d^	118	Boys whose mother participated in the Early Life Exposure in Mexico to Environmental Toxicants (ELEMENT) project	A significant positive association between BPA and SHBG levels (percent change = 4.47; 95% CI: −4.62 to 14.4, *p* = 0.35) and negative association with total (percent change = −17.9; 95% CI: −36.4 to 6.10, *p* = 0.13) and free T levels (percent change = −21.0; 95% CI: −39.7 to 3.31, *p* = 0.09).	Adjusted for urinary specific gravity, child age and child BMI.
Xiong et al. (2015) [22]	Case–control study	China	Controls: 3.8 ± 1.9 Dilated cardiomyopathy patients: 6.9 ± 2.7 ng/mL ^e^	Controls: 59.0 ± 12.7 Dilated cardiomyopathy patients: 59.6 ± 13.2 ^e^	176	Patients diagnosed with dilated cardiomyopathy and age- and gender-matched healthy volunteers	A significant association between serum BPA levels with increased SHBG levels (β = 0.041; 95% CI: 0.024 to 0.058, *p* < 0001).	NA ^c^
Zhou et al. (2013) [23]	Cross-sectional study	China	Exposed: 3.198 Unexposed: 0.276 mg/L ^a^	NA ^c^	290	Male workers exposed to BPA at the workplace	A significant positive association between serum BPA concentration and SHBG level (β = 0.065; 95% CI: 0.009 to 0.120, *p* = 0.023) and inverse associations with androstenedione (β = −0.070; 95% CI: −0.110 to −0.130, *p* = 0.001), free T (β = −0.049; 95% CI: −0.084 to −0.013, *p* = 0.007) and the free androgen index (β = −0.073; 95% CI: −0.130 to −0.016, *p* = 0.012).	Adjusted for age, education, marital status, smoking and alcohol drinking status, history of chronic diseases and medication history.
Galloway et al. (2010) [24]	Cross-sectional study	Italy	3.59 (1.3–11.5) ^b^ ng/mL	20–74 ^d^	715	Italian adults	A significant positive association between BPA excretion and T concentrations in men (β = 0.046; 95% CI: 0.015 to 0.076, *p* = 0.004).	Adjusted for age and study site, and in models additionally adjusted for smoking, measures of obesity and urinary creatinine concentrations.
Lassen et al. (2014) [25]	NA	Denmark	3.25 (0.59–14.89) ^a^ ng/mL	NA ^c^	308	Young men from the general population	A significant positive association between BPA and T (β = 0.7, 95% CI: 0.2 to 1.1, *p* = 0.002), LH (β = 3.5%, 95% CI: −0.02 to 7.1%, *p* = 0.052) and E_2_ (β = 2.7%, 95% CI: 0.4 to 5.1%, *p* = 0.02).	BMI, smoking and time of day of blood sampling.
Mustieles et al. (2017) [26]	Cohort study	Spain	5.1±1.09 ^e^ μg/L	9.8 (9.7, 10.0) ^f^	172	Boys at 9–11 years of age.	A significant association between BPA and higher T levels (β = 1.19, 95% CI = 1.03 to 1.44, *p* = 0.02).	BMT, maternal education, total cholesterol, urinary creatinine and Tanner stage.
Liu et al. (2015) [8]	Cross-sectional study	China	Exposed 685.9 (43.7–3671.8) ^e^ Unexposed 4.2 (0–15.9) μg/g Cr	NA ^d^	592	Male workers from one BPA manufacturer and three epoxy resin manufacturers	A significant association between increasing urine BPA level and increased levels of SHBG (β = 0.0293, *p* = 0.001), E_2_ (β = 0.0362, *p* < 0.001), and a reduced level of FSH (β = 0.0240, *p* = 0.029) and FAI (β = −0.0234, *p* = 0.021).	Age and smoking status.

^a^ Median (5th–95th percentiles); ^b^ Geometric mean (5th–95th percentiles); ^c^ NA, not available; ^d^ Range; ^e^ Geometric mean and standard deviation (GM ± GSD); ^f^ Median (25th–75th percentiles).

**Table 3 toxics-12-00294-t003:** Studies examining the relationship between bisphenol A exposure and semen quality.

Reference	Study Design	Country	BPA Concentration	Age	No. Participants	Population Type	Main Outcome(s)	Adjustments
Goldstone et al. (2015) [27]	Cohort study	USA	0.55 (0.48–0.62) ^a^ ng/mL	31.8 ± 4.9 ^b^	501	Male partners in couples who discontinued contraception	A negative association between BPA and DNA fragmentation (β = −0.0544, *p* = 0.035).	Age, abstinence time, alcohol consumption BMI, creatinine, education, income, previously fathered pregnancy, serum cotinine, study site and race/ethnicity.
Adoamnei et al. (2018) [20]	Cross-sectional study	Spain	2.3 (0.16, 11.5) ^c^ ng/mL	20.4 (18.1–22.8) ^c^	215	Healthy young university students	A negative association between BPA and sperm concentration (β = −0.04, 95% CI: −0.07 to −0.02, *p* < 0.01).	BMI, smoking status, presence of varicocele, urinary creatinine concentration, ejaculation abstinence time and time to start of semen analysis.
Knez et al. (2014) [28]	Cohort study	Slovenia	1.55 (0.3, 6.68) ^d^ ng/mL	34.05 ± 4.76 ^b^	149	Couples undergoing their first or second IVF or intracytoplasmic sperm injection (ICSI) procedure	A negative association between natural logarithm transformed sperm count (β = −0.241, 95% CI: −0.470 to −0.012, *p* = 0.039), perm concentration (β = 0.219, 95% CI: −0.436 to −0.003, *p* = 0.047) and sperm vitality (β = −2.660, 95% CI: −4.991 to −0.329, *p* = 0.026).	Male age, BMI, current smoking status, alcohol consumption and abstinence period.
Mendiola et al. (2010) [11]	Cross-sectional study	USA	1.5 (<LOD–6.5) ^d^ ng/mL	NA ^e^	375	Male partners of pregnant women	No significant associations between urinary BPA concentrations and any of the semen parameters examined.	Age, age squared, BMI, study center, stressful life events and ejaculation abstinence time.
Ji et al. (2018) [29]	Cross-sectional study	China	0.44 ± 5.33 ^b^ μg/g Cr	NA	500	Male couples in less developed areas	A positive association between BPA and linearity (LIN, β = 2.19, 95% CI: 0.37 to 4.0, *p* = 0.0184), straightness (STR, β = 1.47, 95% CI: 0.19 to 2.75, *p* = 0.025), wobble (WOB, β = 1.75, 95% CI: 0.26 to 3.25, *p* = 0.0217), negative correlation with amplitude of lateral head displacement (ALH, β = −0.26, 95% CI: −0.5 to −0.02, *p* = 0.0334) and mean angular displacement (MAD, β = −2.17, 95% CI: −4.22 to −0.11, *p* = 0.0391).	Age, education, race, smoking, alcohol intake, BMI, abstinence period, history of pesticide usage and occupational exposure to high temperature.
Caporossi et al. (2019) [30]	Cross-sectional study	Italy	0.24 ± 0.43 ^b^ μg/g Cr	40.5 (29–67) ^f^	105	Male partners in a fertility clinic	A positive association between BPA and semen volume (β = 0.296, 95% CI: 0.044 to 0.452, *p* = 0.018).	Age, smoke, BMI, alcohol use and genital pathologies
Radwan et al. (2018) [31]	NA	Poland	3.01 ± 5.39 ^b^ μg/L	32.14 ± 4.23 ^b^	315	Men under 45 years of age with normal sperm concentration recruited from a male reproductive health clinic	A negative association between BPA and motility (β = −2.44, 95% CI: −0.26 to 7.48, *p* = 0.03).	Abstinence time, age, smoking, alcohol consumption and past diseases
Lassen et al. (2014) [25]	NA	Denmark	3.25 (0.59–14.89) ^c^ ng/mL	NA	308	Young men from the general population	A negative association between BPA and progressive motile spermatozoa (β = −1.82, 95% CI: −3.10 to −0.53).	BMI, smoking and time of day of blood sampling

^a^ Geometric mean (95% confidence interval); ^b^ Mean ± SD; ^c^ Median (5th–95th percentiles); ^d^ Geometric mean (5th–95th percentiles); ^e^ NA, not available; ^f^ Age (range).

**Table 4 toxics-12-00294-t004:** Studies examining the relationship between bisphenol A exposure and risk of below-reference semen quality parameters.

Reference	Study Design	Country	BPA Concentration	Mean Age ± SD	No. Participants	Population Type	Main Outcome(s)	Adjustments
Meeker et al. (2010) [32]	Cross-sectional study	USA	1.3 (0.8–2.5) ^a^ ng/mL	36.4 ± 4.1 ^b^	190	Male partners of an infertility clinic	A positive association between BPA and ORs for below-reference sperm concentration (OR = 1.47, 95% CI: 0.85 to 2.54), motility (OR = 1.23, 95% CI: 0.83 to 1.80) and morphology (OR = 1.25, 95% CI: 0.77 to 2.06).	Specific gravity, age, BMI, abstinence period, current smoking and time of urine sample.
Chen et al. (2022) [33]	Cross-sectional study	China	2.24 (0.90, 5.30) ^a^	32.0 ± 5.4 ^b^	984	Chinese men from an infertility clinic	A positive association between BPA and ORs of having below-reference sperm concentration (OR = 2.16, 95% CI: 1.13 to 4.13, *p* = 0.04), total sperm count (OR = 2.09, 95% CI: 1.20 to 3.64, *p* = 0.01), progressive motility (OR = 2.09, 95% CI: 1.38 to 3.16, *p* < 0.01) and total motility (OR = 2.09, 95% CI: 1.41 to 3.11, *p* < 0.01).	Age, BMI, abstinence duration, alcohol use, smoking status, education level and ever having fathered a pregnancy.
Pollard et al. (2019) [34]	Cohort study	USA	2.50 (1.81–3.27) ng/mL ^c^	28.5 ± 3.9 ^b^	161	Men ages 18–40 without known subfertility	A positive association between BPA and prevalence ratio (PR) of abnormal sperm morphology (normal heads < 30%) (PR = 1.14, 95%CI: 1.02 to 1.28, *p* = 0.0224) and abnormal sperm morphology (<65% containing normal tails) (PR = 1.13, 95% CI: 0.99 to 1.28, *p* = 0.0081).	Age, race, income, smoking status and BMI
Benson et al. (2021) [35]	Cross-sectional study	Denmark	1.30 (0.22–9.90) ^d^	18–20 ^e^	556	Men from the Fetal Programming of Semen Quality cohort	No associations were observed between urinary BPA concentrations and ejaculate volume, sperm concentration, total sperm count or sperm morphology.	Urine creatinine concentration, alcohol intake, smoking status, BMI, fever, sexual abstinence time, maternal pre-pregnancy BMI and first-trimester smoking, and highest parental education during first trimester.

^a^ Median (25th–75th percentiles); ^b^ Mean ± SD; ^c^ Geometric mean and tertiles; ^d^ Median (5th–95th percentiles); ^e^ Age range.

## Data Availability

The data presented in this study are available on request from the corresponding author.

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
