# Peer review of "Bisphenol A Exposure Interferes with Reproductive Hormones and Decreases Sperm Counts: A Systematic Review and Meta-Analysis of Epidemiological Studies"

_toxics, 2024, doi:10.3390/toxics12040294_

Round 1

Reviewer 1 Report

Comments and Suggestions for Authors

Page 1, Abstract lines 24 and 25 you wrote:

In addition, BPA concentrations were associated with increased sex hormone-binding globulin levels, increased estradiol levels, and reduced biologically…

Then in Page7, you started to define SHBG and estradiol abbreviations. It is better to define them here in the abstract, and then reuse them throughout the manuscript.

In page 7, lines 12 & 13 you wrote: A positive correlation between BPA exposure and sex hormone-binding globulin (SHBG) or estradiol (E2) levels was most frequently. These sentence needs to be mentioned in the abstract as the first definitions to the SHBG and E2. Take the abbreviations between (Brackets) then reuse them throughout.

Page 2, First line 54 the authors wrote:

(The toxicity of BPA is well established at higher doses. At present, measures have been taken to control BPA exposure in affected countries.)

It should read (The toxicity of BPA is well established at higher doses [References are needed here]. At present, measures needs to be replaced with measurements have been taken to control BPA exposure in affected countries [References are needed here].

Final sentence: (The toxicity of BPA is well established at higher doses. At present, measurements have been taken to control BPA exposure in affected countries). Please include the proper references.  

Page 2 line 80 you wrote”

(Medical Subject Headings (MeSH) terms were developed). There is no need to abbreviate the term Medical Subject Headings then take the abbreviation between brackets because you are not going to re-use it again somewhere later. This is the only time you mentioned it. Keep it as is with no abbreviations.

Page 3 Lines 105-111, you wrote:

(The Newcastle-Ottawa Scale (NOS) was used for quality assessment. According to the NOS, each study was awarded a maximum of 9 points, including 4 points for the selection of participants (representation of the exposed cohort, selection of the nonexposed cohort, determination of exposure to BPA, demonstration of the absence of reproductive abnormalities at the start of the study), 2 points for comparability (adjustment for confounding factors, including age, lifestyle, etc.), and 3 points for outcome assessment (relationship between BPA exposure and reproductive hormone levels or sperm parameters as an outcome, enough time for follow-up, adequacy of follow-up of the cohort).)

As you can see, this is not a sentence or a paragraph. This is a run off sentence.

It is a good idea to write short sentences to the point avoiding lengthy sentences or run off sentences.

Lines 112-114 you wrote:

(In the current meta-analysis, studies with a quality score of 7 or higher were considered “high quality”, while others were considered “low quality”. Inconsistent views were resolved through discussions with third parties.)

Now this is very VAGUE Reporting. You need to be clear and to the point by saying while others were considered “low quality” if it score 6 or below such as … and give examples.

In addition, Inconsistent views were resolved through discussions with third parties. Discussion with the third party (WHO? WHERE? and When?)

Page 4 and 5 are Okay

Page 7

All are good except the sentence: A positive correlation between BPA exposure and sex hormone-binding globulin (SHBG) or estradiol (E2) levels… That needs to be cut and pasted in the abstract and the introduction to give the reader what is SHBG stand for before reaching page 7 to find out.

Page 11

All are good.

Page 14

All are good.

Page 17

All are good except you wrote:

(Overall, the results showed that BPA exposure was significantly correlated with sex hormone disturbances since BPA is regarded as a well-known endocrine disrupting chemical [Reference or group of references are needed here]. Its effect on sex hormones has been well established by preclinical studies [Reference or group of references are needed here].)  

The two figures are Okay.

The four tables are Okay. 

References looked good. With exception of:

Ref. 7 you wrote Dose-Response. 2020, 18, where are the page numbers?

Ref, 7 what are these? 1559325820926745.

Ref. 10 (evaluation of the risks to public health related to the presence of bisphenol A (BPA) in foodstuffs. EFSA J. 2023, 21, 06857.) Where are the page numbers?  

                                        What is this number after the volume #21?

Ref. 17 same as above, where are the page numbers?

It is better to go over each reference one-by-one and fix the all author’s names, title of the article, the year of publication, volume #, and insert the correct range of page numbers.

Please review the guide to authors regarding citing the references.

For example, ref. 34 you wrote: I will write the verbatim:

(Benson, T. E.; Gaml-Sørensen, A.; Ernst, A.; Brix, N.; Hougaard, K. S.; Hærvig, K. K.; Ellekilde Bonde, J. P.; Tøttenborg, S. S.; Lindh, C. H.; Ramlau-Hansen, C. H.; Toft, G. Urinary Bisphenol A, F and S Levels and Semen Quality in Young Adult Danish Men. Int J Environ Res Public Health. 2021, 18, 1742. https://doi.org/10.3390/ijerph18041742)

Here you gave everything correct but did NOT:

1-    Give the issue number for volume 18? It should read 18(4), page range-

2-    Give the page range 1742- What?.

Closing points.

I suggest the authors to include the exact name and structure of “Bisphenol A”, because as Scientists/Toxicologists/Chemists the audience of Toxics want to know the structure of the compound that interferes with the reproductive systems in humans/Rodents/mammals.

2,2-Bis(4-hydroxyphenyl) propane CAS 80-05-7 | 803546

In addition, it is a good idea to write short sentences to the point avoiding lengthy sentences or run off sentences.

Best,

Comments on the Quality of English Language

Good English Writings. 

Author Response

We would like to express our sincere thanks to the reviewers for the constructive and positive comments. The manuscript was edited for proper English scientific language by AJE. The comments have been carefully taken into account and a new revised submission has been uploaded. We highlighted all the altered passages in light YELLOW in the tracked copy. The page number of tracked copy was used in the followed responses.

Point-by-point response to Comments and Suggestions for Authors

Comments 1: Abstract lines 24 and 25 you wrote:

In addition, BPA concentrations were associated with increased sex hormone-binding globulin levels, increased estradiol levels, and reduced biologically…

Then in Page7, you started to define SHBG and estradiol abbreviations. It is better to define them here in the abstract, and then reuse them throughout the manuscript.

In page 7, lines 12 & 13 you wrote: A positive correlation between BPA exposure and sex hormone-binding globulin (SHBG) or estradiol (E2) levels was most frequently. These sentence needs to be mentioned in the abstract as the first definitions to the SHBG and E2. Take the abbreviations between (Brackets) then reuse them throughout.

Response 1: Thank you for pointing this out. We agree with this comment. Therefore, we have made changes in Abstract line 26 to insert the abbreviations SHBG and E2 for sex hormone binding proteins and estradiol. And other abbreviated terms were also showed as full name (abbreviation) when first referred, and only the abbreviation was used hereafter.

"In addition, BPA concentrations were associated with increased sex hormone-binding globulin (SHBG) levels, increased estradiol (E2) levels, and reduced biologically active androgen levels."

In page 7, line 166 and 167, we used the abbreviations SHBG and E2 in place of sex hormone binding proteins and estradiol.

"A positive correlation between BPA exposure and SHBG or E2 levels was most frequently observed."

Comments 2: First line 54 the authors wrote:

(The toxicity of BPA is well established at higher doses. At present, measures have been taken to control BPA exposure in affected countries.)

It should read (The toxicity of BPA is well established at higher doses [References are needed here]. At present, measures needs to be replaced with measurements have been taken to control BPA exposure in affected countries [References are needed here].

Final sentence: (The toxicity of BPA is well established at higher doses. At present, measurements have been taken to control BPA exposure in affected countries). Please include the proper references.

Response 2: Thanks so much for your comments. We have revised “measures” to “measurements” to emphasize this point in page 2, line 66. And added appropriate references.

“The toxicity of BPA is well established at higher doses [10]. At present, measurements have been taken to control BPA exposure in affected countries [5, 6].”

Comments 3: Page 2 line 80 you wrote”

(Medical Subject Headings (MeSH) terms were developed). There is no need to abbreviate the term Medical Subject Headings then take the abbreviation between brackets because you are not going to re-use it again somewhere later. This is the only time you mentioned it. Keep it as is with no abbreviations.

Response 3: We are very grateful for your suggestion. In page 3, line 93, we removed the abbreviation MeSH.

“Medical Subject Headings terms were developed by combining subject and free words with “AND” or “OR”: “Bisphenol A”, “male reproduction”, “sperm parameters” and “reproductive hormones” (refer to Table S1)”

Comments 4: Lines 105-111, you wrote:

(The Newcastle-Ottawa Scale (NOS) was used for quality assessment. According to the NOS, each study was awarded a maximum of 9 points, including 4 points for the selection of participants (representation of the exposed cohort, selection of the nonexposed cohort, determination of exposure to BPA, demonstration of the absence of reproductive abnormalities at the start of the study), 2 points for comparability (adjustment for confounding factors, including age, lifestyle, etc.), and 3 points for outcome assessment (relationship between BPA exposure and reproductive hormone levels or sperm parameters as an outcome, enough time for follow-up, adequacy of follow-up of the cohort).)

As you can see, this is not a sentence or a paragraph. This is a run off sentence.

It is a good idea to write short sentences to the point avoiding lengthy sentences or run off sentences.

Response 4: Thank you so much for raising this question. We have amended the paragraph in page 3, line 118 to 124.

“The Newcastle-Ottawa Scale (NOS) was used for quality assessment. According to the NOS, a study can be awarded a maximum of 9 stars. A study fulfills every aspect of the following items get one star. Four stars are set for sample selection, including representativeness of the exposed cohort, selection of the non-exposed cohort, ascertainment of exposure and no outcomes of interest at first. Two stars can be got from comparability whether study was controlled for confounding factors. And 3 stars were set for assessment of outcome, adequacy of follow-up length and adequacy of follow-up cohorts.”

Comments 5: Lines 112-114 you wrote:

(In the current meta-analysis, studies with a quality score of 7 or higher were considered “high quality”, while others were considered “low quality”. Inconsistent views were resolved through discussions with third parties.)

Now this is very VAGUE Reporting. You need to be clear and to the point by saying while others were considered “low quality” if it score 6 or below such as … and give examples.

In addition, Inconsistent views were resolved through discussions with third parties. Discussion with the third party (WHO? WHERE? and When?)

Response 5: Thanks for your comments. We are sorry that we didn't express the meaning of this sentence accurately. What we were trying to say is that a study with a quality score of 6 or below were considered “low quality”. The quality assessment was conducted by two researchers. when two researchers disagree on the quality assessment of the same article, the judgment of the third researcher is used as the final assessment of that article. And the researchers conducted this process were also pointed out in the revised manuscript (page 3, line 124 to 129).

“In the current meta-analysis, studies with a quality score of 7 or higher were considered “high quality”, while that with a quality score of 6 or below were considered “low quality”. Two researchers (Y.L. and S.C.) independently assessed the quality. When two researchers hold in-consistent opinions about an article, the decision of the third researcher (H.X.) is conclusive.”

Comments 6: All are good except the sentence: A positive correlation between BPA exposure and sex hormone-binding globulin (SHBG) or estradiol (E2) levels… That needs to be cut and pasted in the abstract and the introduction to give the reader what is SHBG stand for before reaching page 7 to find out.

Response 6: Thanks for your advice. The full names (abbreviations) of these two terms were applied in the Abstract. In page 7, line 166 and 167, We have used abbreviations for sex hormone-binding globulin or estradiol.

"A positive correlation between BPA exposure and SHBG or E2 levels was most frequently observed."

Comments 7: All are good except you wrote:

(Overall, the results showed that BPA exposure was significantly correlated with sex hormone disturbances since BPA is regarded as a well-known endocrine disrupting chemical [Reference or group of references are needed here]. Its effect on sex hormones has been well established by preclinical studies [Reference or group of references are needed here].)

Response 7: Thank you for your suggestion. We have added appropriate references to increase the reliability of the sentence in page 18, line 224 to 227.

“Overall, the results showed that BPA exposure was significantly correlated with sex hormone disturbances since BPA is regarded as a well-known endocrine disrupting chemical [5, 6]. Its effect on sex hormones has been well established by preclinical studies [23, 24, 26].”

Comments 8: Ref. 7 you wrote Dose-Response. 2020, 18, where are the page numbers?

Ref, 7 what are these? 1559325820926745.

Response 8: We are so sorry for the missing information of the references. We have carefully checked and revised this reference and the others. The page numbers of the Ref. 7 was added in page 19, line 320.

“Cao, T.; Cao, Y.; Wang, H.; Wang, P.; Wang, X.; Niu, H.; Shao, C. The Effect of Exposure to Bisphenol A on Spermatozoon and the Expression of Tight Junction Protein Occludin in Male Mice. Dose-Response. 2020, 18, 1-6. https://doi.org/10.1177/1559325820926745”

Comments 9: Ref. 10 (evaluation of the risks to public health related to the presence of bisphenol A (BPA) in foodstuffs. EFSA J. 2023, 21, 06857.) Where are the page numbers? 

What is this number after the volume #21?

Response 9: Thank you so much for pointing this out. As Ref. 10 were published in electronic journal, only the serial number “e06857” was assigned but not the exact page numbers (page 20, line 327). So the ref. 10 was revised as follow:

“EFSA CEP Panel (EFSA Panel on Food Contact Materials, Enzymes and Processing Aids), Lambré, C.; Barat Baviera, J. M.; Bolognesi, C.; Chesson, A.; Cocconcelli, P. S.; Crebelli, R.; Gott, D. M.; Grob, K.; Lampi, E.; Mengelers, M.; Mortensen, A.; Rivière, G.; Silano, V.; Steffensen, I. L.; Tlustos, C.; Vernis, L.; Zorn, H.; Batke, M.; Bignami, M.; Corsini, E.; FitzGerald, R.; Gundert‐Remy, U.; Halldorsson, T.; Hart, A.; Ntzani, E.; Scanziani, E.; Schroeder, H.; Ulbrich, B.; Waalkens‐Berendsen, D.; Woelfle, D.; Al Harraq, Z.; Baert, K.; Carfì, M.; Castoldi, A. F.; Croera, C.; Van Loveren, H. Re‐evaluation of the risks to public health related to the presence of bisphenol A (BPA) in foodstuffs. EFSA J. 2023, 21, e06857. https://doi.org/10.2903/j.efsa.2023.6857”

Comments 10: Ref. 17 same as above, where are the page numbers? It is better to go over each reference one-by-one and fix the all author’s names, title of the article, the year of publication, volume #, and insert the correct range of page numbers. Please review the guide to authors regarding citing the references.

Response 10: Thank you for your suggestion. Ref.17 is only published online, therefore there is only the reference label without the page number.

Comments 11: Ref. 34 you wrote: I will write the verbatim:

(Benson, T. E.; Gaml-Sørensen, A.; Ernst, A.; Brix, N.; Hougaard, K. S.; Hærvig, K. K.; Ellekilde Bonde, J. P.; Tøttenborg, S. S.; Lindh, C. H.; Ramlau-Hansen, C. H.; Toft, G. Urinary Bisphenol A, F and S Levels and Semen Quality in Young Adult Danish Men. Int J Environ Res Public Health. 2021, 18, 1742. https://doi.org/10.3390/ijerph18041742)

Here you gave everything correct but did NOT:

1-    Give the issue number for volume 18? It should read 18(4), page range-

2-    Give the page range 1742- What?.

Response 11: Thank you for your suggestion. The reference requirement for Toxic journal articles is "Author 1, A.B.; Author 2, C.D. Title of the article. Abbreviated Journal Name Year, Volume, page range. ", with no journal issue number. In order to confirm this doubt, we reviewed recent articles published in the Toxic, none of which were marked with issue numbers in parentheses.

And Ref. 34 was also published on electronic journal, which has no page number range.

Comments 12: I suggest the authors to include the exact name and structure of “Bisphenol A”, because as Scientists/Toxicologists/Chemists the audience of Toxics want to know the structure of the compound that interferes with the reproductive systems in humans/Rodents/mammals.

Response 12: Thank you very much for your suggestion. In order to better explain to the reader, we have added the structural of BPA on page 2, line 57.

Response to Comments on the Quality of English Language

Point 1: In addition, it is a good idea to write short sentences to the point avoiding lengthy sentences or run off sentences.

Response 1: Thank you very much for your suggestion. The original manuscript was edited by native English speaker by AJE on December 2023. In order to make this manuscript easily to read and convey the proper meaning, we tried to replace the long sentence with short sentences in this revision version. The revised sentences were highlighted in page 1, line 36 to 40.

“A number of studies have reported a rising prevalence of infertility with a worsening global fertility rate over the last few decades [1]. A reduced quality of semen caused by physiological and psychological factors individually or in combination substantially contributed to this circumstance [1-3].”

Reviewer 2 Report

Comments and Suggestions for Authors

The manuscript titled „ Bisphenol A Exposure Interferes Reproductive Hormones and 2 Decreases Sperm Counts: A Systematic Review and Meta-Anal- 3 ysis of Epidemiological Studies” focuses on the bisphenol A on male reproductive functions. The manuscript is a very valuable collection of many scientific publications that focus on a very important topic: reproduction. A considerable amount of work has been conducted and with some minor edits and additional information should be acceptable for publication. I would appreciate the authors responding to the following comments.

Comments:

L 35-75 - information about the functioning of the hypothalamic-pituitary-gonadal (HPG) axis should be added in the introduction

L 1-6 page 11 of 21 - the associations between BPA exposure and male semen parameters should be further described

Author Response

We would like to express our sincere thanks to the reviewers for the constructive and positive comments. The comments have been carefully taken into account and a new revised submission has been uploaded. We highlighted all the altered passages in light YELLOW in the tracked copy. The page number of tracked copy was used in the followed responses.

Point-by-point response to Comments and Suggestions for Authors

Comments 1: L 35-75 - information about the functioning of the hypothalamic-pituitary-gonadal (HPG) axis should be added in the introduction.

Response 1: Thank you very much for your comments. We specifically described the action of HPT axis and the effect of BPA on HPT in line 47 to 54, page 2.

“In addition, BPA interferes with the hypothalamic-pituitary-testicular axis where these three glands act in concert to control development, reproduction and aging. Gonadotropin-releasing hormone is secreted from the hypothalamus and stimulate the anterior portion of the pituitary gland to produces gonadotropins luteinizing hormone (LH) and follicle-stimulating hormone (FSH). In response to LH and FSH, the testis produces sex hormone testosterone(T) and inhibit hypothalamus or pituitary gland in return. It was reported that BPA interfered with the synthesis of the FSH, LH, and T levels, and influenced the functional cellular effects of the testes [5, 6].”

Comments 2: L 1-6 page 11 of 21 - the associations between BPA exposure and male semen parameters should be further described.

Response 2: We are very grateful for your suggestion. The detailed association between BPA exposure and male semen parameters was described in line 187 to 193, page 11. The publication biases, the robustness of the results was also described in line 193 to 199, page 11. Besides, we added the description on the study design of included trials (line 182 to 184, page 11). The revised paragraphs were as below:

“In total, 8 studies with 2468 participants reported associations between BPA exposure and sperm parameters, which included sperm concentration, total sperm count, sperm motility, semen volume and normal sperm morphology [11, 20, 26-31]. Of these, four were cross-sectional studies [11, 20, 27, 29], two were cohort studies [28, 30], and the remaining two did not describe the study design [26, 31]. The meta-analysis was conducted according to the β coefficients and 95% confidence intervals of the included studies. The characteristics of these studies are summarized in Table 3.

Round 2

Reviewer 1 Report

Comments and Suggestions for Authors

Keep up the good work.